# Accuracy Assessment in Convolutional Neural Network-Based Deep Learning Remote Sensing Studies—Part 2: Recommendations and Best Practices

Aaron E. Maxwell *,† , Timothy A. Warner † and Luis Andrés Guillén

Department of Geology and Geography, West Virginia University, Morgantown, WV 26505, USA; Tim.Warner@mail.wvu.edu (T.A.W.); lg0018@mix.wvu.edu (L.A.G.)
* Correspondence: Aaron.Maxwell@mail.wvu.edu; Tel.: +1-304-293-2026
† These authors contributed equally to this work.

**Abstract:** Convolutional neural network (CNN)-based deep learning (DL) has a wide variety of applications in the geospatial and remote sensing (RS) sciences, and consequently has been a focus of many recent studies. However, a review of accuracy assessment methods used in recently published RS DL studies, focusing on scene classification, object detection, semantic segmentation, and instance segmentation, indicates that RS DL papers appear to follow an accuracy assessment approach that diverges from that of traditional RS studies. Papers reporting on RS DL studies have largely abandoned traditional RS accuracy assessment terminology; they rarely reported a complete confusion matrix; and sampling designs and analysis protocols generally did not provide a population-based confusion matrix, in which the table entries are estimates of the probabilities of occurrence of the mapped landscape. These issues indicate the need for the RS community to develop guidance on best practices for accuracy assessment for CNN-based DL thematic mapping and object detection. As a first step in that process, we explore key issues, including the observation that accuracy assessments should not be biased by the CNN-based training and inference processes that rely on image chips. Furthermore, accuracy assessments should be consistent with prior recommendations and standards in the field, should support the estimation of a population confusion matrix, and should allow for assessment of model generalization. This paper draws from our review of the RS DL literature and the rich record of traditional remote sensing accuracy assessment research while considering the unique nature of CNN-based deep learning to propose accuracy assessment best practices that use appropriate sampling methods, training and validation data partitioning, assessment metrics, and reporting standards.

**Keywords:** accuracy assessment; thematic mapping; feature extraction; object detection; semantic segmentation; instance segmentation; deep learning





## 1. Introduction

This paper is the second and final component in a series in which we explore accuracy assessment methods used in remote sensing (RS) deep learning (DL) convolutional neural networks (CNN) classification, focusing on scene classification, object detection, semantic segmentation, and instance segmentation tasks. In Part 1 [1], we undertook a review of 100 randomly selected RS DL papers published in 2020 in eight remote sensing journals (*IEEE Geoscience and Remote Sensing Letters*, *IEEE Journal of Selected Topics in Applied Earth Observations and Remote Sensing*, *IEEE Transactions on Geoscience and Remote Sensing*, *International Journal of Remote Sensing*, *ISPRS Journal of Photogrammetry and Remote Sensing*, *Remote Sensing*, *Remote Sensing Letters*, and *Remote Sensing of Environment*). In that review, we found that RS DL studies commonly use metrics and terminology from the computer vision and DL literature, instead of those traditionally used in RS. For example, the class-specific measures of recall and precision are commonly reported instead of the traditional RS equiv-

alents of producer's accuracy (PA) and user's accuracy (UA). The terminology used is, in many cases, inconsistent, with multiple names for similar or identical accuracy measures. For example, we found five other terms sometimes used to describe recall. The studies reviewed rarely reported a complete confusion matrix to describe classification error; and when a confusion matrix was reported, the values for each entry in the table generally did not represent estimation of population properties (i.e., represent a probabilistic sample of the map). Some of these issues are not unique to RS DL studies; similar issues have been noted regarding traditional RS classification accuracy assessment, for example by Foody [2] and Stehman and Foody [3].

Building upon traditional RS accuracy assessment standards and practices, a literature review, and our own experiences, we argue that it is important to revisit accuracy assessment rigor and reporting standards in the context of CNNs. In order to spur the development of community consensus and further discussion, we therefore offer an initial summary of recommendations and best practices for the assessment of DL products to support research, algorithm comparison, and operational mapping. In the Background section, we provide an overview of the types of mapping problems to which CNN-based DL is applicable, review established standards and best practices in remote sensing, and summarize current RS DL accuracy assessment approaches. The Recommendations and Best Practices section outlines accuracy assessment issues related to RS DL classification, focusing on issues related to partitioning the data, assessing generalization, assessment metrics, benchmark datasets, and reporting standards. We end with a discussion of outstanding issues and challenges, along with our overall conclusions.

## 2. Background

### 2.1. CNN and RS Classification

Within geospatial science and RS, CNNs have been applied to a wide range of tasks including classification, spatial predictive modeling, restoration and denoising, pan sharpening, and cloud removal [4–8]. Within the overall task of CNN-based classification, Figure 1 summarizes the four main types of applications. Scene classification is prediction of the category or set of categories that describe an entire image, for example, that the image belongs to the "developed" class (Figure 1a) [4,5,9]. No pixel-level classification or localization is generated. In contrast, object detection includes localization, in which a bounding box with an associated class label and probability is generated [4,5,10]. In Figure 1b, each bounding box represents the location of a building. Semantic segmentation produces a pixel-level classification, similar to that generated by traditional RS methods, where each pixel is assigned to a thematic class, producing a wall-to-wall map (Figure 1c). It is also possible to obtain the associated probabilities for each class for each pixel, information that has applications in spatial probabilistic mapping and modeling [7,8,11]. Lastly, instance segmentation incorporates both object detection and pixel-level classification, with each unique instance of a class delineated, for example, each building in Figure 1d. The potential outputs include bounding boxes, pixel-level masks, predicted classes, and class probabilities [7,8,12,13]. Since the four problem types differ in their application and output, some accuracy assessment best practices will be type-specific.

Table 1 provides some example DL architectures for different mapping problems. DL classification is an active area of research, with new methods and augmented architectures being developed, and thus the examples listed in the table are only a partial list. A full discussion of these techniques can be found in the associated references.

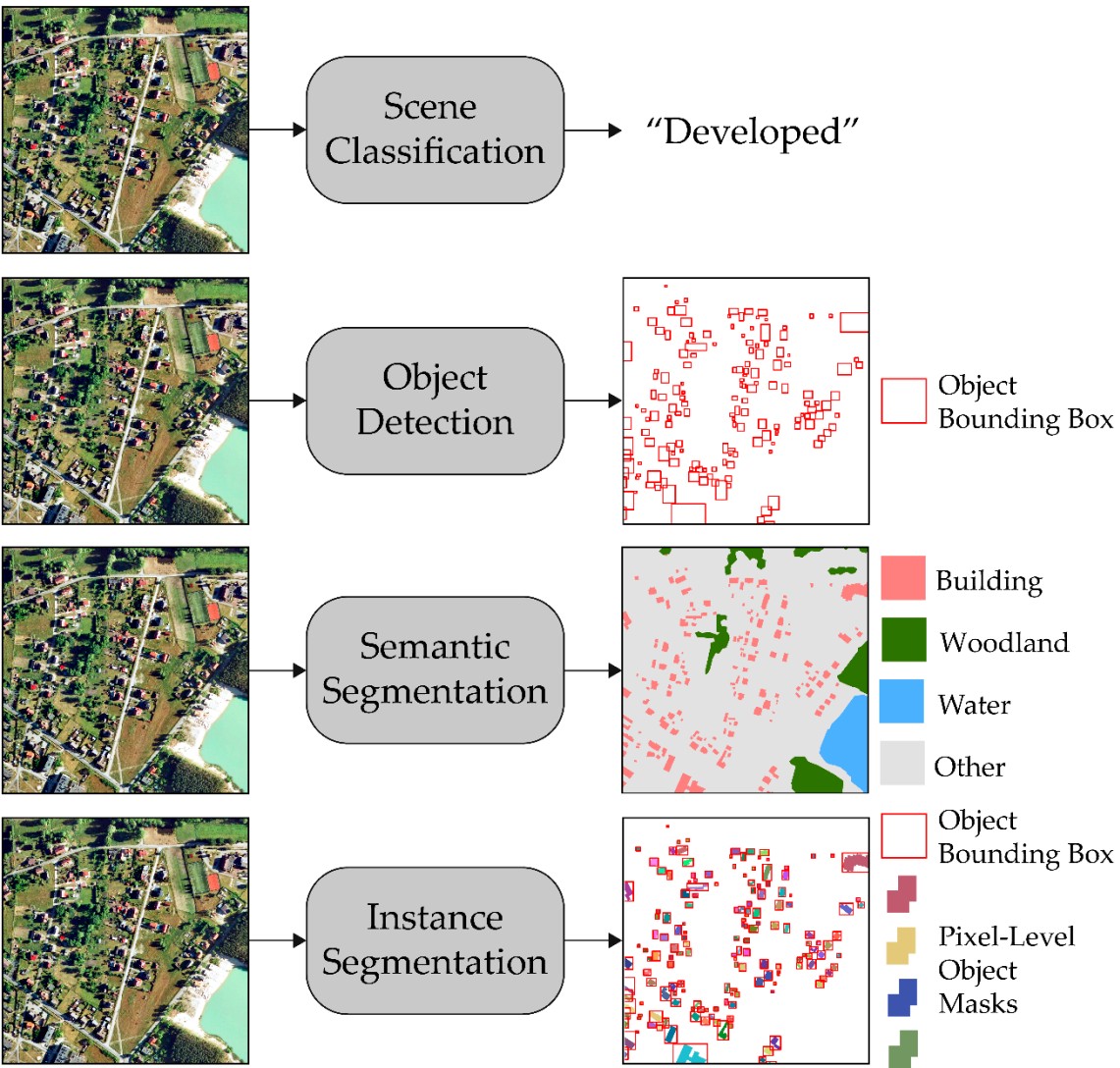

**Figure 1.** Example of different CNN-based mapping problem types. Example data are from the LandCover.ai dataset [14]. (**a**) scene classification, (**b**) object detection, (**c**) semantic segmentation and (**d**) instance segmentation.

**Table 1.** Example DL architectures for the four different types of mapping problems with associated references.

| Group | Examples | Reference |
|---|---|---|
| Scene Classification | AlexNet | [15] |
| | ResNet | [16] |
| | Xception | [17] |
| Object Detection | Single Shot Detector (SSD) | [18] |
| | Yolo v3 | [19] |
| | Faster R-CNN | [20] |
| Semantic Segmentation | SegNet | [21,22] |
| | UNet | [23] |
| | UNet++ | [24,25] |
| | DeepLab | [26] |
| | DeepLabv3+ | [27] |
| Instance Segmentation | DeepMask | [28] |
| | Mask R-CNN | [29,30] |
| | Boundary Preserving Mask R-CNN | [31] |

### 2.2. Traditional Remote Sensing Accuracy Assessment Standards and Best Practices

Decades of RS research and applied studies have yielded accuracy assessment standards and best practices that are generally applicable to a variety of mapping tasks, regardless of the methods applied to obtain results [2,3,32–49]. One of the most important principles is that accuracy should be assessed using randomized, unbiased testing samples that do not overlap with the training samples [37,48]. Despite following the aforementioned recommendations, all datasets have inherent uncertainty and errors, hence the term "ground truth" is usually regarded as misleading; instead, "validation", "testing", and "reference" data are preferred terms. (The distinction between testing and validation data will be discussed in Section 2.3.).

The primary accuracy assessment approach is a spatial overlay of the classification results and testing data, the results of which are summarized on a per-class basis in a table called the confusion matrix. Since the confusion matrix and associated metrics were discussed in detail in Part 1 [1], and comprehensive additional resources are available, for example Congalton and Green [48] and Stehman et al. [37], only a brief overview is provided here.

The confusion matrix is the fundamental source for most derived measures, including overall accuracy (OA), and class-level UA and PA. The goal of generating these accuracy measures is normally to describe the accuracy of map products that represent real landscapes. If the class proportions in the confusion matrix do not reflect the landscape proportions, the error measures derived from that matrix will not reflect the map properties. If a purely random sample is used to create testing samples, then the proportion of samples in each class will be an unbiased estimate of the population properties of the map. However, if other sampling methods are used, such as stratified random sampling, analysis methods that take the sampling design into account should be used to estimate the population proportions from the sample confusion matrix; see for example Stehman's work on this topic [43,50,51].

RS accuracy assessment remains an area of active research, and disagreements on key accuracy assessment issues remain. For example, in geographic object-based image analysis (GEOBIA), the varying size of image objects and associated assessment units has resulted in arguments regarding the appropriate assessment units: pixels, image object polygons, or groups of pixels [45,52–56]. Another area of accuracy assessment research is how to deal with category definitions or transitions between categories that are inherently fuzzy [33,57].

A particularly controversial issue in traditional RS accuracy assessment is the Kappa statistic. Kappa is commonly included in traditional RS accuracy assessment [48], in part because it supposedly provides an adjustment of OA for chance agreement and is robust if class prevalence is imbalanced (i.e., the proportions of the landscape covered by the different classes vary greatly in magnitude) [58]. However, multiple decades of research showing that Kappa is not an unbiased correction of chance agreement, and that it is indeed affected by class imbalance, has resulted in strong recommendations to discontinue the use of Kappa in RS accuracy assessment [59,60].

### 2.3. RS DL Accuracy Assessment Background

As with all machine learning methods, CNN-based classification requires training samples (Figure 2). However, because CNN models include information on spatial context, the input data for CNN models are not individual pixels, as in traditional RS classification, but instead, they are image subsets of a defined size (e.g., 128-by-128, 256-by-256, or 512-by-512 pixels) [7–9,14]. These image subsets, usually called chips or patches, can be generated prior to training and stored on disk, or rendered dynamically during the training process. Since models are trained using image chips, inference also must be performed on individual chips. After the individual chips are classified, they are then usually reassembled to a single, contiguous dataset.

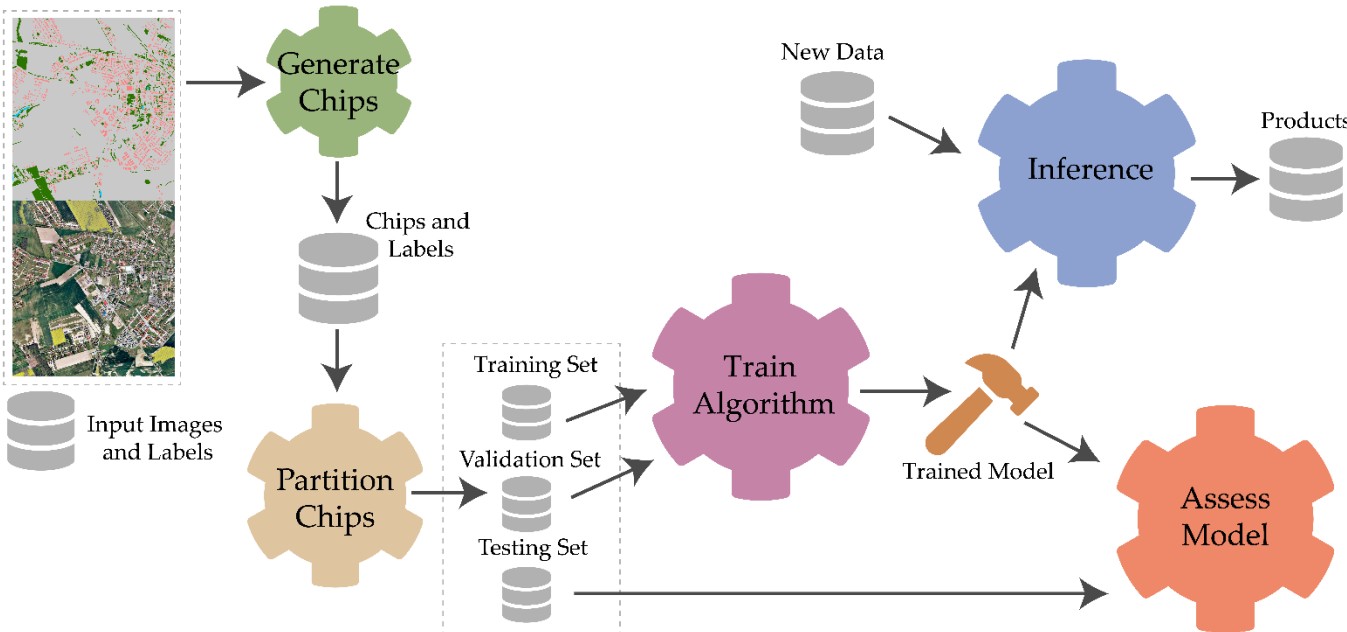

**Figure 2.** Conceptualization of DL workflow including chip generation, data partitioning, training, model assessment, and inference to new data. Example data are from the LandCover.ai dataset [14]. Database and hammer icons are from Font Awesome and made available under a CC by 4.0 license.

CNN-based classifications normally require three sets of reference data, used respectively for training, validation, and testing (Figure 2). The training data are used to empirically determine the parameters of the network, including the spatial filters and weights of the connections. An epoch is one iteration over the entire training dataset. At the end of each epoch, model performance is assessed against the validation image chips, which were not used in training the model, to determine if additional iterations would potentially be useful. The validation dataset provides an independent check for overfitting, a potential problem where the algorithm prioritizes the individual characteristics of the training data over the generalization needed for reliable classification of previously unseen samples. Once a final model is obtained, the testing dataset is used for the final accuracy assessment [7,8,61–63]. Although this terminology of training, validation, and testing is occasionally used in alternative ways, for example referring to the final testing as model validation (e.g., [64]), using these terms in their standard definitions, as indicated above, ensures clarity.

For binary classification specifically, RS DL studies generally use different terms from those used in multi-class problems. However, these metrics are still generated from a confusion matrix. In a binary confusion matrix, the class of interest is referred to as the positive class and the background class as the negative class (Table 2). In a binary classification, four outcomes are possible. True positives (TP) and true negatives (TN) represent the correct classification of respectively the class of interest and the background. A false positive (FP) is the incorrect classification of the background class as the class of interest, whereas a false negative (FN) is the incorrect assignment of the class of interest to the background class. Depending on the nature of the classification, the values reported in the table can potentially represent the fundamental structural units of the dataset (typically pixels, but also occasionally other data structures such as voxels or points in a point cloud dataset), or objects delineated in the dataset. Sometimes the values tabulated represent raw counts, and sometimes they are listed as proportions of each class on the landscape.

The summary accuracy measures typically used for binary DL classifications tend to use unique names for UA and PA for each of the positive and negative classes, with recall and precision being the most commonly used measures (Table 3). Adding further

complexity is that each of the metrics listed in Table 3 has additional names that are sometimes used.

**Table 2.** The binary classification confusion matrix. TP = True Positive, TN = True Negative, FP = False Positive, and FN = False Negative.

|  |  | Reference Data | |
|---|---|---|---|
|  |  | **Positive** | **Negative** |
| Classification Result | Positive | TP | FP |
|  | Negative | FN | TN |

**Table 3.** Binary assessment metrics and relationships to traditional accuracy assessment metrics.

| Metric | Equation | Relation to Traditional RS Measures |
|---|---|---|
| Overall Accuracy (OA) | $\frac{TP+TN}{TP+TN+FP+FN}$ | Overall Accuracy |
| Recall | $\frac{TP}{TP+FN}$ | PA for positives |
| Precision | $\frac{TP}{TP+FP}$ | UA for positives |
| Specificity | $\frac{TN}{TN+FP}$ | PA for negatives |
| Negative Predictive Value (NPV) | $\frac{TN}{TN+FN}$ | UA for negatives |

The F1 score (Equation (1)) is an overall class statistic very commonly used in DL studies and is usually described as the harmonic mean of precision and recall. However, it is also useful to state the equation in terms of the confusion matrix components, using the binary terminology defined in Table 2.

$$\text{F1 Score} = \frac{2 \times TP}{2 \times TP + FN + FP} \tag{1}$$

Another summary statistic often used is the intersection-over-union (IoU) (Equation (2)) [65], which represents the area of intersection divided by the area of union of the reference and predicted class.

$$\text{IoU} = \frac{TP}{TP + FN + FP} \tag{2}$$

As is apparent from comparison of Equations (1) and (2), the F1 score and IoU are strongly correlated. The only difference between the formulas for F1 and IoU is that the latter gives twice the weighting to TP occurrences. When averaged over multiple classes, IoU is typically abbreviated as mIoU. Usually, this averaging is, with each class, equally-weighted. However, a frequency-weighted mIoU (FW mIoU) incorporates the map population characteristics, and therefore is a better representation of the map properties [66].

## 3. Recommendations and Best Practices for RS DL Accuracy Assessment
### 3.1. Training, Validation, and Testing Partitions

The geographic arrangement of the partitioning of the three sets of reference data, used for training, validation, and testing, is important, and should be carried out in a manner that supports the research questions posed [37] and produces unbiased accuracy estimates. Figure 3 summarizes three basic splitting and stratification options for a hypothetical mapping project over the state of Nebraska in the United States (US). Figure 3a

illustrates simple random chip partitioning, in which image chips are created for the entire study area extent, shown in gray, and then these image chips are randomly split into the three partitions without any additional geographic stratification. Figure 3b illustrates geographically stratified chip partitioning in which chips are partitioned into geographically separate regions. Figure 3c illustrates tessellation stratified random sampling, in which the study area is first tessellated into rectangular regions or some other unit of consistent area. Each tessellated region, and all its associated image chips, is then randomly assigned to one of the training, validation, or testing partitions.

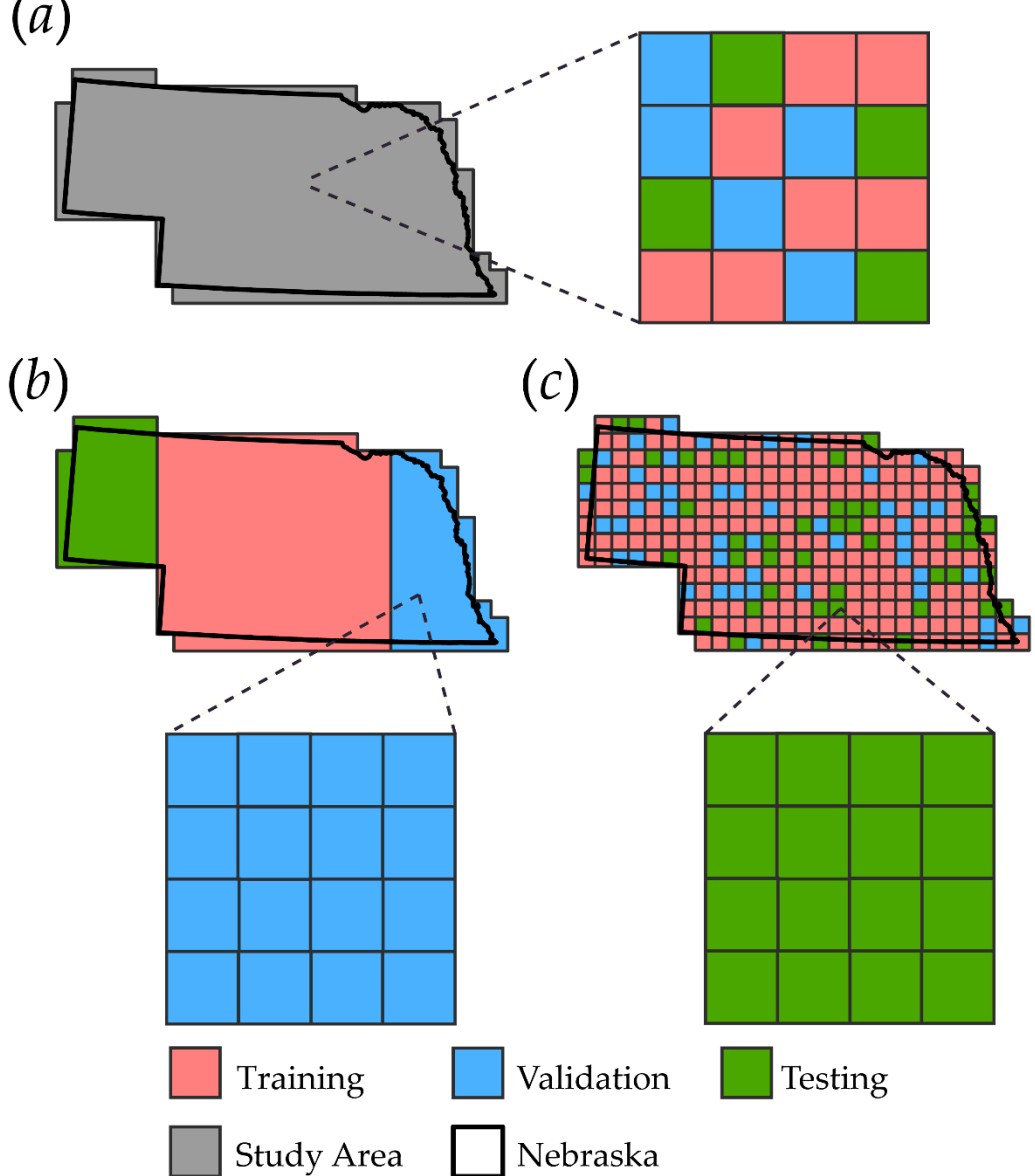

**Figure 3.** Conceptualization of geospatial sampling options used to generate and partition image chips into training, validation, and testing sets. (**a**) Simple random chip partitioning: entire study area extent is divided into chips, which are then randomly split into training, validation, and testing sets. (**b**) Geographically stratified chip partitioning: stratification using geographically contiguous regions. (**c**) Tessellation stratified random sampling: study area extent is first tessellated and then all chips from the same tessellation unit are subsequently randomly assigned to training, validation, and testing sets.

The choice between the three geographic sampling designs described above has potential consequences for accuracy assessment. Both simple random chip partitioning and tessellation stratified random chip partitioning should yield class proportions for each partition similar to those of the entire study area extent. However, if geographically

stratified chip partitioning is used, it is important to bear in mind that the partitioning is normally deliberative and without replication. Thus, before using this approach, one should consider if the study area is uniform, both in terms of the inherent characteristics of the classes, which might affect the classification, and the proportions of the classes in the landscape, which might affect the proportions in the population error matrix. On the other hand, a potential benefit of both the geographically stratified design and the tessellation stratified random sampling design is that the spatial autocorrelation between the partitions is reduced, and thus the accuracy evaluation may be a more robust test of generalization than, for example, simple random chip partitioning. Sometimes subsampling within partitions is also used. If so, whatever the geographic sampling design, the subsampling within the partitions should be probabilistic so that unbiased population statistics can be estimated.

In developing training, validation, and testing datasets, it is also important to consider the impact of overlapping image chips. Many studies have noted reduced inference performance near the edge of image chips [67–70]; therefore, to improve the final map accuracy, it is common to generate image chips that overlap along their edges, and to use only the predictions from the center of each chip in the final, merged output. Overlapping chips are also sometimes generated to increase the number of samples available or represent objects of interest using different positions within the local scene [7,8,68,69]. Generally, the final assessment metrics should be generated from the re-assembled map output, and not the individual chips, since using the chips can result in repeat sampling of the same pixels. Furthermore, if overlap is used when generating chips, all chips that overlap should be maintained in the same data partition so that there is complete independence between the three partitions. For the simple random chip sampling described above, this type of separation may not be possible without subsampling, thus making random chip sampling less attractive.

### 3.2. Assessing Model Generalization

One of the strengths of many RS DL studies is that they explicitly test generalization within their accuracy assessment design, for example by applying the trained model to new data or new geographic regions not previously seen by the model. Conceptually, model generalization assessment is an extension of the geographically stratified chip partitioning method discussed in Section 3.1, except in this case the new data are not necessarily adjacent to or near the training data, and in many cases they involve multiple replications of different datasets. Assessing model generalization is useful both for adding rigor to the accuracy assessment and for providing insight regarding how the model might perform in an operational environment where new training of the model is impractical every time new data are collected. Examples of such generalization tests include Maggiori et al. [71], Robinson et al. [72], and Maxwell et al. [69].

In designing an assessment of model generalization, the type of generalization must be defined, and appropriate testing sets must be developed to address the research question. For example, if the aim is to assess how well a model generalizes to new geographic extents when given comparable data, image chips and reference data in new geographic regions will be needed. If the goal is to assess generalization to new input data, then it may be appropriate to select the new data from the same geographic extent that was used to train the original model.

### 3.3. Assessment Metrics

As highlighted in Part 1 of this study [1], there are many inconsistencies in the RS DL literature relating to which assessment metrics are calculated, how they are reported, and even the names and terminology used. DL RS studies have primarily adopted measures common in the DL and computer vision communities and have abandoned traditional RS accuracy assessment metrics and terminology. We argue here that when choosing which metrics to calculate and report, the use and purpose of the derived products should be

considered. It is important to report metrics that give end users and other researchers insight into how the model and dataset will perform for the application of interest. It is also important to consider the relative proportion of classes or features of interest on the landscape to estimate accuracy metrics that reflect the population from which the samples were drawn. Below, we recommend best practices for choosing and reporting accuracy assessment metrics for the four CNN-based problem types with these considerations in mind.

### 3.3.1. Scene Classification

In scene classification, the unit of assessment is a single image or chip. Though the image chips are usually generated from geospatial data, the classified images are not normally mapped or referenced back to a specific spatial location in the assessment. This lack of a map output might suggest that the concept of the population confusion matrix is not relevant. Nevertheless, potentially operationalizing the method implies a specific population from which the input data will be drawn. Therefore, as with any classification, the class prevalence data in the confusion matrix should match that of the assumed population. In practice, however, it may be challenging to generate a population matrix for scene classification. Many scene classification benchmark datasets appear to be deliberative samples, and usually by design collect samples that are not in proportion to the likelihood of the class in the population.

An example of a scene classification dataset is DeepSat SAT-6 [9] (available from: https://csc.lsu.edu/~saikat/deepsat/; accessed on 1 July 2021). This dataset differentiates six classes (barren, building, grassland, road, tree, and water) and consists of 28-by-28 pixel image chips derived from National Agriculture Imagery Program (NAIP) orthophotography. A total of 405,000 chips are provided, with 324,000 used for training and 81,000 reserved for testing, as defined by the dataset originators. Figure 4 provides some example image chips included in this dataset.

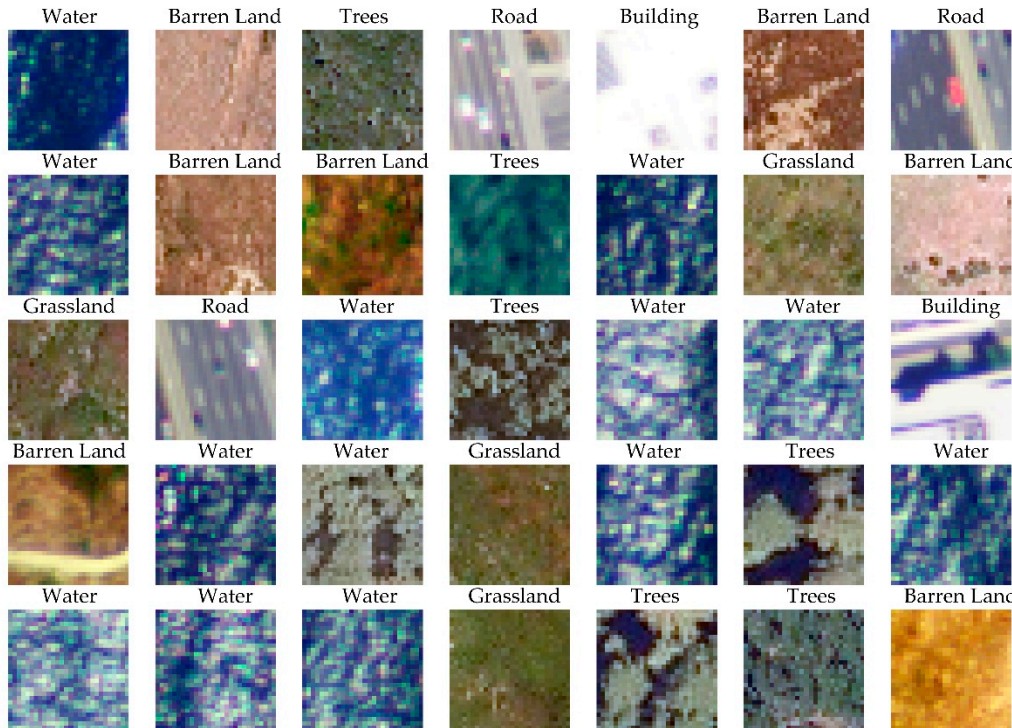

**Figure 4.** Example DeepSat SAT-6 28-by-28 pixel image chips with associated labels provided by [9].

Table 4 summarizes the accuracy of a ResNet-32 CNN-based scene classification in terms of the confusion matrix with values in the table representing the proportions of

the classes in the original testing dataset. Thus, the row and column totals represent the prevalence of each class in the classified data and reference data, respectively. Reporting the entire confusion matrix, along with row and column totals, as well as the class accuracy statistics is useful because it allows greater understanding of how the statistics were calculated, as well as the different components of the classification errors. We use DL terms of precision and recall, as well as the traditional RS terms of UA and PA for clarity.

**Table 4.** Scene labelling accuracy assessment for DeepSat SAT-6 dataset, with values in the table representing proportions of outcomes based on class prevalence in the benchmark dataset. OA = 0.9967.

| | | Reference | | | | | | Row Total | Precision (UA) | F1 |
|---|---|---|---|---|---|---|---|---|---|---|
| | | Barren | Building | Grassland | Road | Tree | Water | | | |
| Classification | Barren | 0.2248 | 0.0000 | 0.0006 | 0.0000 | 0.0000 | 0.0000 | 0.2253 | 0.9974 | 0.9943 |
| | Building | 0.0000 | 0.0454 | 0.0000 | 0.0000 | 0.0000 | 0.0000 | 0.0455 | 0.9989 | 0.9946 |
| | Grassland | 0.0020 | 0.0000 | 0.1548 | 0.0000 | 0.0002 | 0.0000 | 0.1570 | 0.9861 | 0.9909 |
| | Road | 0.0000 | 0.0004 | 0.0000 | 0.0255 | 0.0000 | 0.0000 | 0.0260 | 0.9819 | 0.9899 |
| | Tree | 0.0000 | 0.0000 | 0.0001 | 0.0000 | 0.1749 | 0.0000 | 0.1750 | 0.9996 | 0.9993 |
| | Water | 0.0000 | 0.0000 | 0.0000 | 0.0000 | 0.0000 | 0.3712 | 0.3712 | 1.0000 | 1.0000 |
| **Column Total** | | 0.2268 | 0.0459 | 0.1555 | 0.0256 | 0.1751 | 0.3712 | | | |
| **Recall (PA)** | | 0.9912 | 0.9903 | 0.9957 | 0.9981 | 0.9989 | 1.0000 | | | |

Table 5 provides the same data, except this time the columns are normalized to sum to 1.0 (see for example [73]). Because each class has the same column total, this confusion matrix represents a hypothetical case where each class has the same prevalence.

**Table 5.** Scene labelling accuracy assessment for DeepSat SAT-6 dataset, with accuracy values in each column summing to 1.0, an approach often used in DL studies. OA = 0.9957.

| | | Reference | | | | | | Row Total | Precision (UA) | F1 |
|---|---|---|---|---|---|---|---|---|---|---|
| | | Barren | Building | Grassland | Road | Tree | Water | | | |
| Classification | Barren | 0.9912 | 0.0000 | 0.0037 | 0.0000 | 0.0000 | 0.0000 | 0.9949 | 0.9962 | 0.9937 |
| | Building | 0.0000 | 0.9903 | 0.0000 | 0.0019 | 0.0000 | 0.0000 | 0.9922 | 0.9981 | 0.9942 |
| | Grassland | 0.0088 | 0.0000 | 0.9957 | 0.0000 | 0.0011 | 0.0000 | 1.0056 | 0.9902 | 0.9929 |
| | Road | 0.0000 | 0.0097 | 0.0002 | 0.9981 | 0.0000 | 0.0000 | 1.0079 | 0.9902 | 0.9941 |
| | Tree | 0.0000 | 0.0000 | 0.0004 | 0.0000 | 0.9989 | 0.0000 | 0.9993 | 0.9996 | 0.9993 |
| | Water | 0.0000 | 0.0000 | 0.0000 | 0.0000 | 0.0000 | 1.0000 | 1.0000 | 1.0000 | 1.0000 |
| **Column Total** | | 1.0000 | 1.0000 | 1.0000 | 1.0000 | 1.0000 | 1.0000 | | | |
| **Recall (PA)** | | 0.9912 | 0.9903 | 0.9957 | 0.9981 | 0.9989 | 1.0000 | | | |

Designers of community benchmark datasets, including SAT-6, sometimes do not make it clear whether the proportions of the samples in the various classes represent the prevalence of those classes in the landscape. Thus, it is not clear if Table 4 is truly a population estimate. However, to illustrate how a population estimate can be obtained from these data, we assumed an application in the East Coast of the USA, and obtained a high spatial resolution (1 m$^2$ pixels) map of the 262,358 km$^2$ Chesapeake Bay region from [74]. In Table 6, the values in the table have been normalized so that the column totals represent the class prevalence values determined from the Chesapeake reference map, and thus, unlike the other two tables, Table 6 provides an estimate of a potential real-world application of the dataset.

**Table 6.** Scene labelling accuracy assessment for DeepSat SAT-6 dataset, with class prevalence set to equal the estimated prevalence in the Chesapeake Bay region obtained from [74], and thus the confusion matrix represents an estimate of the population matrix. OA = 0.9978.

| | | Reference | | | | | | Row Total | Precision (UA) | F1 |
|---|---|---|---|---|---|---|---|---|---|---|
| | | **Barren** | **Building** | **Grassland** | **Road** | **Tree** | **Water** | | | |
| Classification | Barren | 0.0033 | 0.0000 | 0.0010 | 0.0000 | 0.0000 | 0.0000 | 0.0043 | 0.7647 | 0.8633 |
| | Building | 0.0000 | 0.0277 | 0.0000 | 0.0000 | 0.0000 | 0.0000 | 0.0277 | 0.9989 | 0.9946 |
| | Grassland | 0.0000 | 0.0000 | 0.2715 | 0.0000 | 0.0007 | 0.0000 | 0.2721 | 0.9975 | 0.9966 |
| | Road | 0.0000 | 0.0003 | 0.0000 | 0.0157 | 0.0000 | 0.0000 | 0.0160 | 0.9803 | 0.9891 |
| | Tree | 0.0000 | 0.0000 | 0.0001 | 0.0000 | 0.6199 | 0.0000 | 0.6200 | 0.9998 | 0.9994 |
| | Water | 0.0000 | 0.0000 | 0.0000 | 0.0000 | 0.0000 | 0.0598 | 0.0598 | 1.0000 | 1.0000 |
| | **Column Total** | 0.0033 | 0.0280 | 0.2726 | 0.0157 | 0.6206 | 0.0598 | | | |
| | **Recall (PA)** | 0.9912 | 0.9903 | 0.9957 | 0.9981 | 0.9989 | 1.0000 | | | |

Comparing Tables 4–6, it is notable that the values for recall are the same in each table, but precision, F1, and OA are different. This emphasizes that class prevalence affects most summary accuracy metrics, and therefore the prevalence values used are important. Assuming all classes have equal prevalence, as in Table 5, appears to be the standard for RS scene classification. In our survey of 100 papers, five of the 12 studies that dealt with scene classification reported confusion matrices, and all five used this normalization method. However, a hypothetical landscape in which all classes exist in equal proportions is likely to be rare, if it is found at all. It is also important to note that, since the number of classes determines the prevalence of each class in the hypothetical equal-prevalence landscape, such an approach does not facilitate comparisons between studies, unless the studies all have the same number of classes. Kappa, though only occasionally used in DL studies, is sometimes also suggested as useful if the data are imbalanced, but as discussed previously should be avoided [59,60].

As an alternative, some studies only report recall, on the basis that this metric is not affected by prevalence [65,75]. However, Table 6 highlights the potential hazard of such an approach: the classification has values for recall (PA) above 0.99 for all classes, but the barren class has a precision (UA) value of only 0.76. If only recall values are tabulated, the user would be misled as to the reliability of the classification for barren.

### 3.3.2. Semantic Segmentation

The typical output of a DL semantic segmentation is a wall-to-wall, pixel-level, multi-class classification, similar to that of traditional RS classification. For such classified maps, traditional RS accuracy assessment methods, including using a probabilistic sampling scheme to facilitate unbiased estimates of population-based accuracies, and reporting a complete confusion matrix, has direct application. Using traditional terminology, such as UA and PA, would facilitate communication with RS readers, but almost all authors seem to prioritize communication with the DL community and use the computer science or computer vision terms, such as precision and recall. Given that there are so many of these alternative names for the standard RS metrics, perhaps the most important issue is that all metrics should be clearly defined.

Table 7 gives an example confusion matrix for a semantic segmentation, and Table 8 lists the number of samples used in calculating the accuracy measures, emphasizing the large size of the assessment sample. To produce this table, the LandCover.ai multiclass dataset [14] (available from https://landcover.ai/; accessed on 1 July 2021) was classified with an augmented UNet architecture [23], using the training and testing partitions defined by the originators. In Table 7, the values are proportions of the various combinations of reference and classified classes in the landscape, and the column totals represent the

class prevalence values. For example, Buildings is a rare class, making up just 0.8% of the reference landscape.

**Table 7.** Confusion matrix for semantic segmentation of LandCover.ai dataset. Values in the table represent estimates of the population (i.e., landscape) proportions. OA = 95.0%.

| | | Reference | | | | Row Total | Precision (UA) | F1 Score |
|---|---|---|---|---|---|---|---|---|
| | | **Buildings** | **Woodlands** | **Water** | **Other** | | | |
| Classification | Buildings | 0.007 | 0.000 | 0.000 | 0.003 | 0.009 | 0.827 | 0.716 |
| | Woodlands | 0.000 | 0.325 | 0.000 | 0.019 | 0.344 | 0.931 | 0.937 |
| | Water | 0.000 | 0.001 | 0.053 | 0.004 | 0.058 | 0.961 | 0.938 |
| | Other | 0.001 | 0.023 | 0.002 | 0.562 | 0.588 | 0.956 | 0.955 |
| | **Column Total** | 0.008 | 0.349 | 0.056 | 0.588 | | | |
| | **Recall (PA)** | 0.705 | 0.944 | 0.916 | 0.955 | | | |

**Table 8.** Accuracy assessment data used for semantic segmentation of LandCover.ai dataset.

| Class | Sample Size (No. Pixels) |
|---|---|
| Buildings | 246,966 |
| Woodlands | 9,035,022 |
| Water | 1,531,777 |
| Other | 15,433,404 |

Table 7 illustrates why the F1 score on its own, without the values of the constituent precision and recall measures, provides only limited information. The F1 scores of Woodlands and Water are almost identical, suggesting the classification performance is basically the same for these two classes. Table 7, however, shows that Water, unlike Woodlands, had a much lower recall than precision.

Surprisingly, although our review of 100 DL papers found the use of many different accuracy metrics, none used some of the more recently developed RS metrics, such as quantity disagreement (QD) and allocation disagreement (AD) suggested by Pontius and Millones [59], which provide useful information regarding the different components of error. For the map summarized in Table 7, the QD was 0.0% and the AD was 5.0%, indicating almost no error is derived from an incorrect estimation of the proportion of the classes; instead, the error derives from the mislabeling of pixels.

Table 9 was derived from [68], which used semantic segmentation to extract the extent of surface mining from historic topographic maps. The accuracy assessment included a geographic generalization component, and was carried out on the mosaicked images, not individual image chips. The results in Table 9 highlight the value of reporting a variety of metrics. Due to the large area of the background or "not mining" class, the OA and specificity were very high. However, variability in the accuracy of mapping the mining classes and issues of FPs and FNs were captured by precision, recall, and the F1 score. Reporting only OA would have been misleading in this case, due to the class imbalance. Reporting only the F1 score would obscure the fact that in Geographic Region 4, the much lower accuracy is due to a low recall, whereas the precision is similar to that of the other areas.

**Table 9.** Binary classification accuracy assessment from prior study [68] in which mining features were extracted from historic topographic maps. NPV = Negative Predictive Value.

| Geographic Region | Precision | Recall | Specificity | NPV | F1 Score | OA |
|:---:|:---:|:---:|:---:|:---:|:---:|:---:|
| 1 | 0.914 | 0.938 | 0.999 | 0.993 | 0.919 | 0.999 |
| 2 | 0.883 | 0.915 | 0.999 | 0.999 | 0.896 | 0.998 |
| 3 | 0.905 | 0.811 | 0.998 | 0.993 | 0.837 | 0.992 |
| 4 | 0.910 | 0.683 | 0.998 | 9.983 | 0.761 | 0.983 |

CNN-based deep learning can also be used to generate spatial probabilistic models. For example, Gagne et al. [76] explored DL for the probabilistic prediction of severe hailstorms, while Thi Ngo et al. [77] assessed landslide susceptibility modeling. If the primary output will be a probabilistic model, probabilistic-based assessment metrics should be reported. The main probabilistic-assessment metrics are areas under the curve (AUC) derived from the receiver operating characteristic (ROC) and the precision-recall (P-R) curves; however, if the end user will primarily make use of the "hard" classification, then threshold-based derived metrics should also be reported [78,79]. A disadvantage with the AUC ROC metric is that it relies on recall and specificity, and thus only the producer's accuracies for the positive and negative classes are considered [65,79]. Thus, precision (i.e., UA) is not assessed, which can be especially misleading when class proportions are imbalanced. As a result, the P-R curve and associated AUC PR metric is more commonly recommended. This topic is explored in more detail in the context of object detection in Section 3.3.3.

### 3.3.3. Object Detection and Instance Segmentation

Object detection and instance segmentation both identify individual objects. Since the number of true negative objects (i.e., objects not part of the category of interest) is generally not defined, reporting a full confusion matrix and the associated OA metric is generally not possible. Nevertheless, reporting the number of TP, TN, FN, and FP, along with the derived accuracy measures typically used—precision, recall, and F1 (e.g., [80])—ensures clarity in the results.

A further complication for object detection is that there is normally a range of confidence levels in the detection of any single object. This confidence potentially has two components: the overall probability that an object exists, and the degree to which the identified object is correctly delineated. The first of these components can be represented by the P-R curve and associated AUC metric, as discussed in Section 3.3.2. In the context of object detection, however, the AUC PR is normally referred to as average precision (AP), and/or mean average precision (mAP). The second component of object probability, the delineation of the object, is usually quantified in terms of the IoU of the reference and predicted masks or bounding boxes. For example, the threshold IoU value of 0.5 is often used to determine whether a sufficient number of pixels are correctly labeled for an object to be regarded as correct (i.e., a TP); otherwise it is labeled either a FN or FP. However, because the choice of a threshold is usually arbitrary, a range of thresholds may be chosen, for example, from 0.50 to 0.95, with steps of 0.05, which would generate 10 sets of object detections, each with its own P-R curve and AP metric.

Unfortunately, there is considerable confusion regarding the AP/mAP terminology. First, these terms may be confused with the average of the precision values in a multiclass classification (e.g. a semantic segmentation). Second, mAP is generally used to indicate a composite AP, for example, typically averaged over different classes [81–83], but also sometimes different IoU values [84]. However, because of inconsistencies in the usage of the terms AP and mAP in the past, some sources (e.g., [85]) no longer recommend differentiating between them, and instead use the terms interchangeably.

Because of the lack of standardization in AP and mAP terminology, it is important for accuracy assessments to clearly report how these accuracy metrics were calculated, for example, specifying the IoU thresholds used. Adding subscripts for IoU thresholds,

if thresholds are used, or BB or M to indicate whether the metrics are based on bounding boxes or pixel-level masks, can be an effective way to communicate this information (for example, $AP_{BB}$ or $IoU_M$). Wu et al. [86] used both superscripts and subscripts to differentiate six AP metrics derived from bounding box and pixel-level masks, as well as three different sizes of objects (small, medium, and large).

In reporting object-level data, it is important to consider whether a count of features or the area of features is of interest. For example, a model may be used to predict the number of individual trees in an image extent. In such a case, it would be appropriate to use the numbers of individuals as the measurement unit. However, if the aim is to estimate the area covered by tree canopy, then it would be more appropriate to use assessment metrics that incorporate the relative area of the bounding boxes or masks.

### 3.4. Creating Benchmark Datasets

A variety of benchmark geospatial datasets are available to test new methods and compare models. For a detailed list of public benchmark DL datasets for use in RS, see Hoeser and Kuenzer [8]. Cheng et al. [87] provide an extensive comparison of benchmark datasets. Despite the range of publicly available benchmark datasets, there are notable gaps in the available data. For example, datasets to test generalization to new geographic extents and/or data are limited, and there is also generally a lack of non-image benchmark datasets, such as those representing LiDAR point clouds, historic cartographic maps, and digital terrain data, as noted in our prior studies [68,69]. Cheng et al. [87] argue that the differentiation of a larger number of classes is particularly important.

Benchmark datasets and the associated metadata inherently constrain the potential for rigorous accuracy assessment, depending on the nature of the data and, crucially, how they are documented. Benchmark datasets in many cases provide pre-determined protocols for accuracy assessment. Therefore, it is particularly helpful if benchmark datasets support best practices, as described below:

1.  The methods used for generating training, validation, and testing partitions should be documented.
2.  Tables and/or code should be provided to replicate the defined data partitions. Alternatively, the data partitions can be separated into different folders.
3.  Class codes and descriptions should be defined.
4.  The testing dataset should approximate the relative proportions of classes or features on the landscape in order to support the generation of population-level metrics. Sampling should either be comprehensive (i.e., wall-to-wall) or probabilistic.
5.  If the dataset is meant to support the assessment of model generalization to new data and/or geographic extents, the recommended validation configuration should be documented.
6.  The accuracy of the reference data should be assessed and reported.
7.  Example code should be provided as a benchmark.
8.  Any complexities or limitations of the dataset should be defined.

### 3.5. Reporting Standards

In order to improve consistency between studies, replication capacity, and inference potential, it is important to report, in detail, the research methods and accuracy assessment metrics. Here, we provide a list of reporting standards and considerations that will promote clarity for interpretation and replication of experiments.

1.  The number of training, validation, and testing chips should be reported, along with the chip size, classes and associated numeric codes, class definitions, and means of partitioning the samples (i.e., simple random, geographically stratified, tessellation stratified random, etc.). The relative proportions of each mapped category or abundance of features of interest on the actual landscape should be described.
2.  If a subset of image chips from the entire dataset and/or mapped extent is used, the method for this subsampling should be described.

3. If the validation/testing sample unit was not the image chip (i.e., pixels, pixel blocks, or areal units), the unit should be stated and explained. If a minimum mapping unit (MMU) is used, this should be stated and the rationale for its use documented.

4. Providing code and data on a public access site is particularly helpful. If doing so, this should include code or a description that allows the data partitions used in the original study to be replicated. For example, providing code that will replicate the partitions when executed, and/or specifying random seeds could be used to enhance replicability.

5. The supporting documentation or code should allow the generation of a confusion matrix and associated derived metrics that are estimates of the population characteristics. If the testing sample does not directly provide an estimate of the population properties, methods for estimating the population confusion matrix from the sample confusion matrix should be provided.

6. Any thresholds used to generate metrics should be reported (e.g., IoU thresholds or probability thresholds to generate binary metrics).

7. For object detection and/or instance segmentation, it should be stated whether bounding boxes and/or pixel-level masks were used to perform the assessment, and whether assessment relied on feature counts or areas.

8. If model generalization is assessed, the form of generalization should be defined along with the sampling methods, geographic extents, and/or new datasets used.

9. Any limitations of the input data should be described.

## 4. Outstanding Issues and Challenges

Although CNN-based deep learning has shown great promise in geospatial object detection and thematic mapping, there are some issues and complexities that hinder rigorous, systematic experimentation and application, which should be discussed. First, comparison of new methods and architectures is hindered by the complexity of hyperparameter optimization and architecture design [4,5,7,8]. When experimenting with traditional machine learning methods, it is generally possible to optimize hyperparameters using a grid search and cross validation to compare a variety of settings [88]. However, due to lengthy training times, computational demands, and a wide variety of hyperparameters and architecture changes that can be assessed, such systematic experimentation is currently not possible for CNN-based DL [4,5,7,8,89]. When new algorithms or methods are proposed, it is common to use existing machine learning or DL methods as a benchmark for comparison; however, we argue that such comparisons are limited due to an inability to optimize the benchmark model fully [72,90]. For example, Musgrave et al. [90] suggest that reported improvements when using new algorithms and architectures are often overly optimistic due to inconsistencies in the input data, training process, and experimental designs and because the algorithm to which the comparison is being made was not fully optimized. Fairer and more rigorous comparisons generally suggest more marginal improvements.

This issue is further confounded by the lack of a theoretical understanding of the data abstractions and generalizations modeled by CNNs [91]. For example, textural measures, such as those derived from the gray level co-occurrence matrix (GLCM) after Haralick [92], have shown varying levels of usefulness for different mapping tasks; however, one determinant of the value of these measures is whether or not the classes of interest have unique spatial context or textural characteristics that allow for greater separation [93–96]. Building on existing literature, a better understanding of how CNNs represent textural and spatial context information at different scales could potentially improve our general understanding of how best to represent and model textural characteristics to enhance a wide range of mapping tasks. Furthermore, improved understanding could guide the development of new DL methods and architectures.

Accuracy estimates are usually empirical estimates of map properties, and as with all statistical estimates, they have uncertainties. In the RS DL 100 papers surveyed in our literature review [1], only a small subset provided quantitative estimates of uncertainty.

Following the recommendation of Musgrave et al. [90], we suggest that confidence intervals be reported where possible. For example, confidence intervals can be estimated for overall accuracy [97] and AUC ROC [98] or multiple model runs can be averaged to assess for variability. For example, Oh et al. [82] use the mean and standard deviation of 30 replications to assess variability. It is also possible to statistically compare output. For example, McNemar's test offers a statistical comparison of two classifications [32] and the Delong test can compare AUC ROC values [99]. Such information can be informative, especially when only small differences in assessment metrics between models are observed.

The rich history of RS accuracy assessment research and application offers recommendations that are applicable to thematic mapping tasks broadly, including those relying on DL. For example, accuracy assessment methods generally assume that all pixels will map perfectly to the defined classes and that feature boundaries are well-defined. However, real landscapes are much more complex, and some thematic transitions, such as the transition between uplands and wetlands [100–102], are generally gradational rather than discrete. Foody [33] notes the harshness of standard accuracy assessment methods relative to these issues. In a prior study [57], we outline an assessment approach based on weighting the center of reference and predicted features higher than areas near boundaries as a more appropriate assessment of accuracy when boundaries are inherently fuzzy, such as wetlands and individual tree crowns. A wide variety of assessment metrics can be generated using center-weighted measures, including metrics that are straight forward to interpret in that they are similar to conventional measures (e.g., center-weighted OA, precision, recall, and F1 score). The impact of landscape complexity and heterogeneity should also be considered when designing accuracy assessment protocols and interpreting results. For example, heterogenous landscapes may prove more difficult to map in comparison to more homogeneous landscapes, resulting from more boundaries, edges, class transitions, and, potentially, mixed pixels or gradational boundaries [33,103]. Accounting for mixed pixels, gradational boundaries, difficult to define thematic classes, and the defined minimal mapping unit (MMU) are complexities that exist for all thematic mapping tasks, including those relying on DL, which highlight the need for detailed reporting of sample collection and accuracy assessment methods [1–3,32,33,44,45,57,104,105].

Researchers should also consider the regional, biome, and landscape-specific mapping needs and complexities presented in the existing literature. For example, prior studies offer insights specific to land cover and vegetation mapping in Mediterranean biomes (e.g., [106–109]), northern tundra and permafrost regions (e.g., [110–112]), rangeland (e.g., [113,114]), and urbanized areas (e.g., [115–117]). Such findings can inform DL researchers as to existing mapping difficulties and research needs, which classes are most difficult to differentiate, and which classes are most important to map for different use cases. We argue that considering the existing literature, methods, and complexities relating to specific mapping tasks or landscape types can offer guidance for knowledge gaps and research needs to which DL may be applicable. Similarly, the current body of knowledge offers insights for extracting information from specific data types. For example, extracting information from true color or color infrared aerial imagery with high spatial resolution, limited spectral resolution, and variability in illuminating conditions between images is an active area of research in the field [14,71,118–121].

## 5. Conclusions

Building on a previous review of 100 RS DL papers [1], we propose best practices for the assessment of a wide variety of thematic, geospatial products derived using CNN-based DL. Since RS DL studies generally use geospatial data, many of the findings of traditional accuracy assessment methods have direct application in RS DL studies, and are indeed reflected in current RS DL accuracy assessments, as shown by the literature review. For example, most (but not all) RS DL studies seem to be heeding the general consensus that Kappa is a potentially misleading statistic and so should be avoided.

However, RS DL studies appear to depart from traditional RS terminology in notable ways. Because the new terminology has not been entirely standardized, it is useful if studies define the metrics used, and provide the underlying equations. For example, for metrics such as AP and mAP, threshold(s), type of mask and any subdivision by object size should preferably be indicated with superscripts and/or subscripts. Another important issue is that relatively few RS DL studies report full confusion matrices, and those that do, generally do not report values in the table that are estimates of population proportions. Confusion matrices with class prevalence values that do not reflect the map or landscape classified do not provide predictions that can be linked to actual applications.

Some aspects of RS DL analysis require accuracy assessment protocols that are designed for unique issues associated with CNN-based classification. The typically overlapping nature of the image chips that are the standard input for DL classification requires special consideration for designing training, validation, and testing partitions. The training, validation, and testing partitions should be entirely separate, even when chip overlap regions are taken into account. Classification should be assessed based on the reassembled image, rather than the individual chips. Model generalization is a particularly useful aspect of many RS DL accuracy assessments. By incorporating multiple generalizations, studies can both improve the robustness of the accuracy test and provide insight regarding the likely performance of the model in real-world situations where the model may need to use new data without additional extensive training.

Considering the significant increase in the application of DL methods to RS applications in recent years, and none withstanding the fast advancement of this field, it is crucial to carefully consider how accuracy assessments are carried out. Hence, classification products should be assessed in a manner that quantifies their accuracy relative to an intended use and that takes into account the geographic nature of the map product, which is an approximation of an actual landscape. Rigorous assessment should be supported by clear reporting that promotes transparency and reproducibility. Adoption of the best practices outlined here, and refinement of these suggestions in the future, will foster rigor and consistency in geospatial DL experimentation.

**Author Contributions:** Conceptualization, A.E.M. and T.A.W.; formal analysis, A.E.M., T.A.W., and L.A.G.; investigation, A.E.M., T.A.W., and L.A.G.; data curation, A.E.M.; writing—original draft preparation, A.E.M. and T.A.W.; writing—review and editing, A.E.M., T.A.W., and L.A.G. All authors have read and agreed to the published version of the manuscript.

**Funding:** This work was funded by the National Science Foundation (NSF) (Federal Award ID Number 2046059: "CAREER: Mapping Anthropocene Geomorphology with Deep Learning, Big Data Spatial Analytics, and LiDAR").

**Institutional Review Board Statement:** Not Applicable.

**Informed Consent Statement:** Not Applicable.

**Data Availability Statement:** Not Applicable.

**Acknowledgments:** We would like to acknowledge all originators of the public datasets used in this study including the United States Geological Survey (USGS), West Virginia View, Louisiana State University, the Chesapeake Conservancy, and the Poland Head Office of Geodesy and Cartography. We would also like to thank three anonymous reviewers whose comments strengthened the work.

**Conflicts of Interest:** The authors declare no conflict of interest.

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
