# Peer review of "Accuracy Assessment in Convolutional Neural Network-Based Deep Learning Remote Sensing Studies—Part 2: Recommendations and Best Practices"

_remotesensing, doi:10.3390/rs13132591_

Round 1
Reviewer 1 Report
The accuracy assessment in Convolutional Neural Network-Based deep learning remote sensing studies is an interesting topic but the paper seems to be an old cliché rather than the inspiring idea. There are several issues. I'd like to discuss with the authors:
- The accuracy assessment of deep learning-based techniques at the regional scale is a little bit over-researched. The author should provide a detailed explanation about deep learning-based and traditional remote sensing techniques specifically at complex terrain region is more significant and urgently needed.
- Multispectral satellite imagery with three channels, preferably true-color (red, green, and blue) or false-color (green, red, near infrared) composite with radiometric depth of 8 bit or 16 bit have a great impact in DL based techniques. It can be R,G,B or G,R,NIR but you dint introduce any recent studies.
- Consideration of contextual information (e.g. edges, vegetation, shape, area, and the consistency of feature distributions) increases the reliability of the DL model classification -- Not clear where and how it was used at all. For the review manuscript, the author should discuss this issue.
- None of the results have been discussed in terms of tundra types. Please provide more DL-based breakdown works for tundra types to understand the model performance besides just an observation.
- There are lots of practical challenges that apply to very high spatial resolution (VHSR) multispectral (MS) commercial satellite imagery over aerial imagery. Any Recommendations for best Practices? Also, multispectral images are always partitioned into regular patches with smaller sizes and then individually fed into deep neural networks (DNNs) for semantic segmentation. If these images are independent of one another in terms of geographic spatial information, what would be the challenge and recommendation?
- In terms of Scene/object classification, it will be great if you add below provided studies:
Abdulla, W. Mask R-Cnn for Object Detection and Instance Segmentation on Keras and Tensorflow. Available online: https://github.com/matterport/Mask_RCNN (accessed on 25 August 2020).
Dai et el. 2016. Instance-aware semantic segmentation via multi-task network cascades. In Proceedings of the 2016 IEEE Conference on Computer Vision and Pattern Recognition, Las Vegas, NV, USA, 27–30 June 2016;IEEE: Piscataway, NJ, USA, 2016.
Ren, et el. 2016. Three-dimensional object detection and layout prediction using clouds of oriented gradients. In Proceedings of the IEEE Conference on Computer Vision and Pattern Recognition, Las Vegas,NV, USA, 27–30 June 2016; pp. 1525–1533.
Witharana, et el. 2020. Understanding the synergies of deep learning and data fusion of multispectral and panchromatic high resolution commercial satellite imagery for automated ice-wedge polygon detection. ISPRS J. Photogramm. Remote Sens. 2020, 170, 174–191.
Li, et el. 2016:Fully convolutional instance-aware semantic segmentation. arXiv 2016, arXiv:1611.07709.
Wei, X.; Fu, K.; Gao, X.; Yan, M.; Sun, X.; Chen, K.; Sun, H. Semantic pixel labelling in remote sensing images using a deep convolutional encoder-decoder model. Remote Sens. Lett. 2018, 9, 199–208.
Reviewer 2 Report
In this review, the authors build on a previous review of 100 RS DL papers (Maxwell et al., in review), and propose best practices for the assessment of a wide variety of thematic, geospatial products derived using CNN-based DL. The main content of deep learning method which used in remote sensing areas is properly analyzed. I have no additional comments and I recommend to publish it as it
Reviewer 3 Report
The work has merit, and this reviewers can get the novelty.
There is clear the goal of the research work.
description of wide span tractor and their influence is sufficient for
research.
Round 2
Reviewer 1 Report
Many thanks to the authors for carefully considering my feedback to improve the manuscript.